# Microsatellite Instability and Aberrant Pre-mRNA Splicing: How Intimate Is It?

**DOI:** 10.3390/genes14020311

**Published:** 2023-01-25

**Authors:** Laurent Corcos, Enora Le Scanf, Gaël Quéré, Danielle Arzur, Gwennina Cueff, Catherine Le Jossic-Corcos, Cédric Le Maréchal

**Affiliations:** 1Inserm U1078, Univ Brest, EFS, F-29200 Brest, France; 2CHRU Brest, F-29200 Brest, France

**Keywords:** pre-mRNA splicing, digestive cancers, microsatellite instability, DNA replication errors, DNA damage

## Abstract

Cancers that belong to the microsatellite instability (MSI) class can account for up to 15% of all cancers of the digestive tract. These cancers are characterized by inactivation, through the mutation or epigenetic silencing of one or several genes from the DNA MisMatch Repair (MMR) machinery, including *MLH1*, *MLH3*, *MSH2*, *MSH3*, *MSH6*, *PMS1*, *PMS2* and *Exo1*. The unrepaired DNA replication errors turn into mutations at several thousand sites that contain repetitive sequences, mainly mono- or dinucleotides, and some of them are related to Lynch syndrome, a predisposition condition linked to a germline mutation in one of these genes. In addition, some mutations shortening the microsatellite (MS) stretch could occur in the 3′-intronic regions, i.e., in the *ATM* (ATM serine/threonine kinase), *MRE11* (MRE11 homolog) or the *HSP110* (Heat shock protein family H) genes. In these three cases, aberrant pre-mRNA splicing was observed, and it was characterized by the occurrence of selective exon skipping in mature mRNAs. Because both the *ATM* and *MRE11* genes, which as act as players in the MNR (*MRE11*/*NBS1* (Nibrin)/RAD50 (RAD50 double strand break repair protein) DNA damage repair system, participate in double strand breaks (DSB) repair, their frequent splicing alterations in MSI cancers lead to impaired activity. This reveals the existence of a functional link between the MMR/DSB repair systems and the pre-mRNA splicing machinery, the diverted function of which is the consequence of mutations in the MS sequences.

## Foreword

This paper is presented as a hypothesis article. We sincerely apologize to the authors whose contributions to the field have not been discussed.

## 1. A Quick Look at the Pre-mRNA Splicing Unit (Figure 1)

Pre-mRNA splicing, the excision of introns and ligation of exons, is an obligatory step towards protein synthesis for most human genes. It is orchestrated by the spliceosome, a macromolecular complex with above 150 distinct proteins and several small nuclear RNAs. The spliceosome acts through repeated cycles of subunits recruitment that interact with specific RNA sequence determinants, the 5′ donor site (GU mainly), the 3′ acceptor site (AG, almost exclusively), a polypyrimidine-enriched sequence (PPT) of at least 7–8 nucleotides immediately upstream of the 3′ intronic splice site, and the branch point (BP), a sequence centered around a given adenosine (A) residue that lies upstream from the PPT, roughly between the −17 and −50 intronic positions relative to the 3′ splice site [1]. Several proteins interact with these sequence elements, including U2AF1 (U2 small nuclear RNA auxiliary factor 1) onto the AG dinucleotide at the 3′ splice site, U2AF2 (U2 small nuclear RNA auxiliary factor 2) onto the PPT, SF1 (Splicing factor 1) and SF3B1 (Splicing factor 3b subunit 1) onto the BP and proteins from the U1 spliceosome sub-unit onto the 5′ splice site [2,3]. Other proteins from the SR-SF (serine-arginine rich-splicing factors) or from the heterogenous nuclear RNA-binding proteins (hnRNP) families participate in splicing decisions upon interacting with either intronic or exonic sequences, and they enhance or suppress selected splicing events [1,4]. Pre-mRNA splicing occurs constitutively with marked tissue specificity and according to specific cell requirements, for instance, at various developmental stages or in an adaptive mode in response to environmental constraints, such as exposure to drugs [5,6]. Indeed, it has been recently proposed that splicing events that are characteristic of human early development are reactivated in tissue-specific cancers [7]. These alternative splicing programs govern the synthesis of specific proteins with neo-antigen potential, although some of the mRNAs are too short lived to encode proteins, as they are degraded according to the nonsense-mediated mRNA decay (NMD) control mechanism that senses the occurrence of premature STOP codons resulting from frame-disrupting pre-mRNA sequence alterations [8]. Importantly, pre-mRNA splicing and transcription are coupled, such that when RNA polymerase II proceeds at a fast pace, the introns with suboptimal sites may not be recognized efficiently, resulting in exon skipping, which has consequences, as mentioned above [9,10]. Importantly, it has also become quite clear that pre-mRNA splicing is profoundly modified in pre-cancerous as well as in cancerous lesions [3,11]. This results from both *cis*-occurring events, such as mutations in protein binding sites, or from *trans*-occurring events, such as mutations in RNA-binding proteins [12]. Among the splice-altering mutations, a specific case can be made of those involving DNA replication errors, such as PPT alterations that are believed to result from the slippage of the DNA polymerase [13,14]. These changes mostly happen as the PPT shortens, thereby minimizing the interaction of U2AF2 with its cognate pre-mRNA target sequence. This is a hallmark of Microsatellite Instability (MSI) cancers, which are defective in the activity of the MMR pathway because of the silencing, by mutations or epigenetic mechanisms, of some of the genes involved in repairing DNA replication errors [15]. With this in mind, altered pre-mRNA splicing has been proposed both as a biomarker of diseases or as a target to innovative therapeutic strategies. For instance, the BcL-x (BcL2L1, B-cell lymphoma 2L1) pre-mRNA is spliced into two transcripts-encoding proteins with opposite functions in apoptosis. Hence, the long BcL-xL isoform prevents apoptosis, while the short BcL-xS isoform promotes apoptosis [16,17]. Several approaches have been devised to selectively deprive cancer cells of the expression of BcL-xL, including the use of splice-switching oligonucleotides that mask the usage of the BcL-xL splice site, resulting in the re-sensitization of the cancer cells to chemotherapy [18,19]. In addition, some candidate drugs have shown promising effects in various preclinical trials, as well as in clinical trials, however, with marked side effects [20]. In addition, many drugs that disrupt the spliceosome function have been discovered [21], including pladienolide, which targets U2-associated SF3B1 and showed important selectivity against cancer cells. In colon cancer, PRPF6 (Pre-mRNA processing factor 6), a member of the tri-snRNP (small nuclear ribonucleic particle) spliceosomal complex, was shown to drive cancer cell proliferation by the preferential splicing of genes associated with growth regulation, making it a plausible target [22]. Several additional splice-corrective approaches have been devised [23].

**Figure 1 genes-14-00311-f001:**
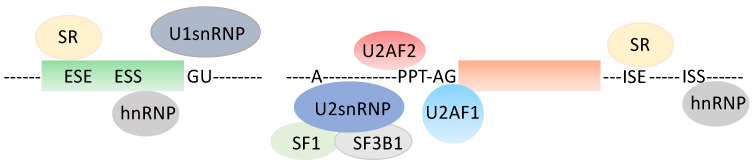
Generic splice unit. ESE: Exonic Splice Enhancer, ESS: Exonic Splice Silencer, ISE: Intronic Splice Enhancer, ISS: Intronic Splice Silencer, PPT: polypyrimidine tract, hnRNP: heterogenous nuclear ribonucleic particle, snRNP: small nuclear ribonucleic particle, SR: serine arginine-rich RNA-binding protein.

## 2. Microsatellite Instability Cancers and Pre-mRNA Splicing Alterations (Table 1, Figure 2)

Microsatellite (MS) sequences, repetitions of mono- or di-nucleotides, occur throughout the genome, with a higher proportion of them being in introns than in exons [24,25,26]. These repetitive elements are particularly prone to acquiring mutations, resulting from the slippage of the DNA polymerase, i.e., where the correct nucleotide has been inserted, but the sequence prior to it has slipped. The mismatch repair (MMR) system comprises many proteins, including MLH1 (MutL homolog 1), MLH3 (MutL homolog 3), MSH2 (MutS homolog 2), MSH3 (MutS homolog 3), MSH6 (MutS homolog 6), PMS1 (PMS1 homolog 1), PMS2 (PMS1 homolog 2) and EXO1 (Exonuclease 1) [27]. It safeguards the genome from the occurrence of detrimental variations, but it becomes poorly efficient in MSI cancers, a frequent class of cancers with an instable genome that account for up to 15% of cancers from the digestive tract [28]. The MS instability, which is characterized by reductions or, more rarely, increases in the nucleotide repetitions, is a consequence of inactivating mutations or epigenetic silencing of the *PMS2* or *MLH1* MMR genes, respectively. Individuals with Lynch syndrome (HNPCC, Hereditary Nonpolyposis Colorectal Cancer) carry a germline pathogenic variant in one of their MMR genes, which makes them highly susceptible to developing digestive cancers when the secondary mutation or epigenetic silencing of an MMR gene takes place [29]. Pre-mRNA alterations, such as those occurring in intronic PPT, are highly specific for MSI cancers, while they happen very occasionally in Microsatellite Stable (MSS) cancers [30]. Historically, the consequences of intronic MS modifications have not been fully appreciated, but they have been linked to changes in the splicing patterns, namely, increased skipping of the exon that follows the intron being affected [31]. This correlated with the impairment of the Double Strand Break (DSB) repair system. This DNA damage sensing system is mobilized in cases of DNA replication and stress-associated DNA double-strand breaks [32]. These are induced by oxidations resulting from an altered metabolism or in response to DNA damaging agents such as chemotherapeutic drugs or in cases of R-loops formation that leave single-stranded DNA highly vulnerable to damage [33]. In the *ATM*, *MRE11* and *HSP110* genes, the exon skipping phenomenon is always detected at given frequencies, leading to various degrees of impairment of the protein activity or in the production of a chaperone Hsp110 isoform, permitting better responses to 5-fluorouracile in this subset of colon cancer patients [34,35]. It should be also mentioned that MSI colon cancers are endowed with an overall better prognosis than that of MSS cancers, and they respond better to immune check point inhibitors [36,37]. Gastric cancers also show MSI-associated alterations, including in intron 4 of the *MRE11* gene [31], and many additional splice site mutations accompanied by aberrant splicing products were also identified, including in the *FHIT* (Fragile Histidine Triad Diadenosine Triphosphatase) gene, a putative tumor suppressor gene [38]. The impact of the mutations in the *ATM* and the *MRE11* genes, which are thought to participate in resolving DSBs, was investigated.

**Table 1 genes-14-00311-t001:** Intronic PPT in *ATM*, *MRE11* and *HSP110* genes. The intronic splice acceptor site (**ag**) is preceded by the PPT. GC: gastric cancer, CRC: colorectal cancer. WT: Wild Type.

Gene	Ensembl Transcript	Skipped Exon N°	WT Intronic 3′ End
*ATM*	ENST00000675843.1	6 (GC)	**ctaataatttttttttttttttaag**
*ATM*	ENST00000675843.1	12 (CRC)	**tgaagctttttgtttttctttgtag**
*MRE11*	ENST00000323929.8	5	**ttttaagtaactttttttttttaag**
*HSP110*	ENST00000320027.10	9	**catgatttttttttttttttttaag**

**Figure 2 genes-14-00311-f002:**
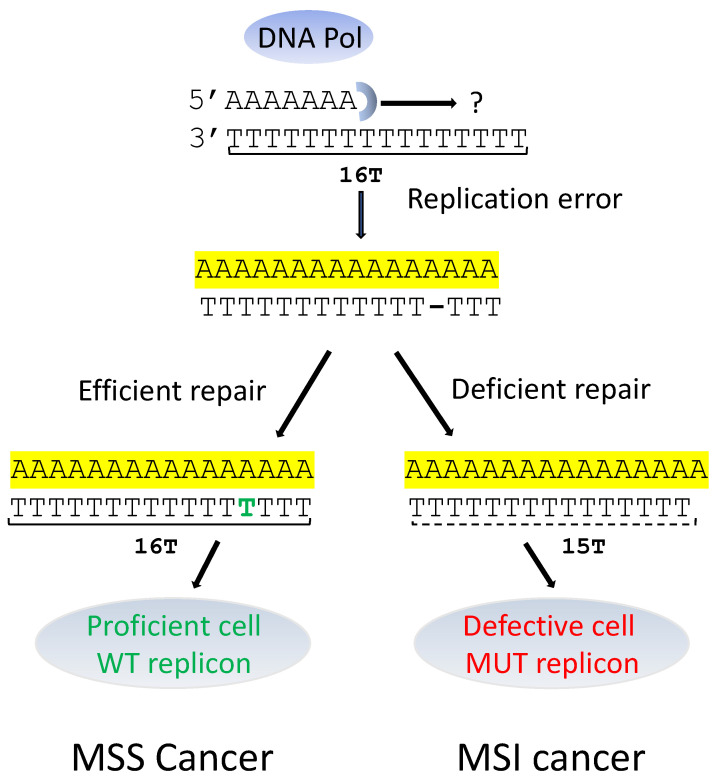
MSS and MSI splicing in cases of DNA replication errors. WT: Wild Type; MUT: Mutant. DNA replication errors may be fixed (MSS cancer) or not fixed (MSI cancer).

## 3. Activation of the MMR and DSB Repair Mechanisms (Table 2)

The type of cell response to DNA damage specifies the Chromosomal Instability (CIN) or MSI, depending on activity of the MMR genes. The MMR system corrects the DNA replication errors, while DNA DSB are repaired through homologous recombination and non-homologous end joining instead [32]. However, because the *MRE11* and *ATM* genes may undergo specific intronic PPT changes in MSI cancers, resulting in alterations of their splicing, and thus, their activity, we hypothesize that in at least some MSI cancers, the DSB repair machinery is less active, which may result in the worsening of the deleterious effects of the DNA damage. Even though both the *ATM* and *MRE11* genes are overexpressed in the MSI cell lines, the cell cannot cope with the impactful splice alterations elicited by the altered PPT, especially considering that the vast majority of PPT mutations are biallelic. Indeed, it was shown that MSI cancer cells could no longer efficiently repair DSBs in the context of splice-altering mutations in the *ATM* gene or in the *MRE11* gene from the MNR (*MRE11/NBS1*/RAD50) DNA mutation repair system [33,39]. In MSI cancers, the *MRE11* gene may show a 1 or 2 pyrimidine deletion in the PPT from intron 4. This leads to impaired activity of the MNR complex because of a loss of activity of MRE11. Importantly, the biallelic mutations of MRE11 account for roughly 39% of MSI cancers, and across 70 MSI primary human cancers, MRE11 mutations occurred in 83.7 and 50% of colorectal and endometrial cancers, respectively [31]. Other genes from the MNR complex involved in DNA repair, such as *ATR*, do not carry any mutation in the intronic PPT. Overall, DSB repair machinery genes, including in the *DNAPKcs*, *RAD50*, and *BRCA1* genes, are frequently mutated in mononucleotide repetitions in MSI endometrial cancers [40]. In addition, a rapid search in the TCGA database for cancer-specific mutations in regions that lie within or very close to *cis*-splice sites have revealed the occurrence of many other genes with intronic PPT mutations occurring in various tissues with MSI cancers. How RNA-binding proteins are central to the DSB repair programs has been recently analyzed in depth [41].

**Table 2 genes-14-00311-t002:** Frequent PPT mutations in and across various MSI cancers. Mutations occurring at or very near RNA-binding sites of splice proteins involved in splicing were retrieved from the TCGA/ICGC database. The table shows the names of the genes, their associated mutation numbers, and the numbers of occurrence in the groups of cancers. LMAN1 (Lectin, mannose binding 1) (e) is an exonic mutation, while LMAN1(Lectin, mannose binding 1) (i) is an intronic mutation.

		CDC14	ADAM28	PAMR1	XYLT2	DNAJC18	FRMD4	ATL3	DDX6	FHOD3	ZBTB20	PSME4	GSE1	LARP4B	SCLO6A1	ZNF43	PREPL	RNF43	LMAN1 (e)	LMAN1 (i)
**Primary site**	N	MU138468	MU103116	MU1871022	MU62168	MU158136	MU61476	MU71434	MU69837	MU59026	MU92398	MU56855	MU81682	MU89465	MU73674	MU57884	MU93009	MU70114	MU5428109	MU91855
Colon polyps	**66**	2	4	4	5	7	7	9	9	11	12	13	14	14	16	17	24	38	40	44
Colon	**5**	3							2			2			1					1
Stomach	**5**		1		1	1	1	2	1	1			1	1	1	2	2	3	3	4
Genital tractus	**14**		1		1	2				1	1	2		1		6	1	2	1	6
Others (skin, brain)	**5**							2			1	1	2		1	1	1			
	**N = 95**																			

## 4. Crossroads between DNA Repair and Pre-mRNA Splicing Programs

Ataxia Telangiectasia is a recessive multiorgan genetic disorder that is caused by mutations in the *ATM* gene [42]. ATM is activated in response to DSB, which triggers the phosphorylation of many proteins involved in DNA repair, cell cycle, or apoptosis [43]. It may also be activated in response to other types of stresses, such as those caused by DNA topoisomerase cleavage complexes or R-loops [44]. In MSI cancers, *ATM*’s intronic MS (a stretch of 15 T) shows a 4-nucleotide homozygous deletion (or hemizygous deletion + loss of heterozygosity), resulting in a decrease in the ATM protein level. Importantly, ATM participates in regulating the activity of the U2AF38 *Drosophila Melanogaster* splicing factor following ionizing radiation, and it controls UV light-associated spliceosome displacement, a phenomenon that is linked to increased R-loop formation [45]. The level of phosphorylation of SRSF1, an accessory splicing factor that mainly stimulates exon inclusion, is potentially afforded by the serine/threonine kinase activity of ATM. In an *ATM*-mutated context, SRSF1 phosphorylation, and hence, its activity, could be modified in response to exposure to DNA-damaging agents [46]. Furthermore, several members of the SR protein family (SRSF1, SRSF2, and SRSF3) can prevent the formation of R-loops that may result from splicing reactions [47,48]. HnRNP, the polypyrimidine tract-binding protein-associated splicing factor (PSF) or the RBMX protein rapidly accumulate at sites of DNA damage [49]. PRPF19 (Pre-mRNA processing factor 19), a protein from the core of the catalytically active spliceosomes [50], is strongly up-regulated in response to DNA damage, while its down-regulation results in DSBs after exposure to irradiation. In addition, Prp19 is phosphorylated by ATM in response to oxidative stress and DSB-inducing agents [51]. It was also shown that RBM14 (RNA-Binding Motif Protein 14), another splicing protein, is recruited to DSB sites, and BRCA1 (Breast Cancer gene 1), which is involved in cell cycle check point and homologous recombination, interacts with the splicing factor BCLAF1 (BCL2-Associated Transcription Factor 1), leading to alterations of the pre-mRNA splicing of several DNA damage response genes, including *ATRIP* (ATR interacting protein), *BACH1* (BTB and CNC Homology 1) and *EXOI* [52]. Hence, several players in the splicing machinery are both modified and mobilized by DNA damage. Although some of this evidence is indirect, these observations suggest, collectively, that the modifications to the level and/or localization of RNA-binding proteins bridge pre-mRNA splicing and DNA damage. Whether all of this applies to MSI cancers as well is unknown. Nevertheless, the extent to which this inter-dependency between the splicing and the DNA damage repair machineries shapes the cells’ adaptive responses needs to be fully explored. Already, according to our current knowledge, it may be that their reciprocal links play a major role against the risks of corrupted gene expression programs.

## 5. Ever-Growing Interplay between Pre-mRNA Splicing and the DNA Damage Response (Figure 3)

We present here data illustrating the relationship between two major cell machineries, DNA repair and pre-mRNA splicing. We describe how such systems, in concert, contribute to adapting cell responses to cancer triggers. MSI cancers are characterized by frequent mutations in the intronic PPT, essentially reducing the numbers of pyrimidine nucleotides that lead to the unbalanced splicing associated with increased exon skipping. Remarkedly, exon skipping occurs perfectly accurately from the first to the last nucleotide of the next exon. This suggests that the spliceosome can no longer properly recognize the intron–exon junction [53]. Our previous [9] studies lend support to this hypothesis. Because intronic PPTs are mandatory for the splicing reaction to proceed properly, their integrity has to be maintained, coping with the constraints that dictate the maintenance of large variability in the sequence and position relative to the 3′AG intronic splice acceptor site. Importantly, PPTs are also present in exons, with a similar propensity to acquire cancer-linked mutations. How PPT mutations are selected in MSI cancers remains a largely open question, but it may be plausible that an overall remodeling of the splicing decisions, such as it occurs in MSI cancers, may initiate early on in the course of cancer progression. This hypothesis is supported by our identification of recurrent MS mutations in MSI early cancers, such as colon adenomas (Table 2). More work is needed to obtain a complete picture of the interactions between the DNA replication errors and damage and pre-mRNA splicing, and potentially, protein synthesis. It is unclear to what extent these mutations can be repaired in MSI cancers, but in case no repair can be achieved, there still is the possibility in a fraction of cases that the exon-lacking mRNAs can be eliminated through the activation of the NMD pathway, an mRNA quality control that participates in dampening the effects of the numerous types of genomic DNA aggressions. A model presenting our working hypothesis is presented in Figure 3.

**Figure 3 genes-14-00311-f003:**
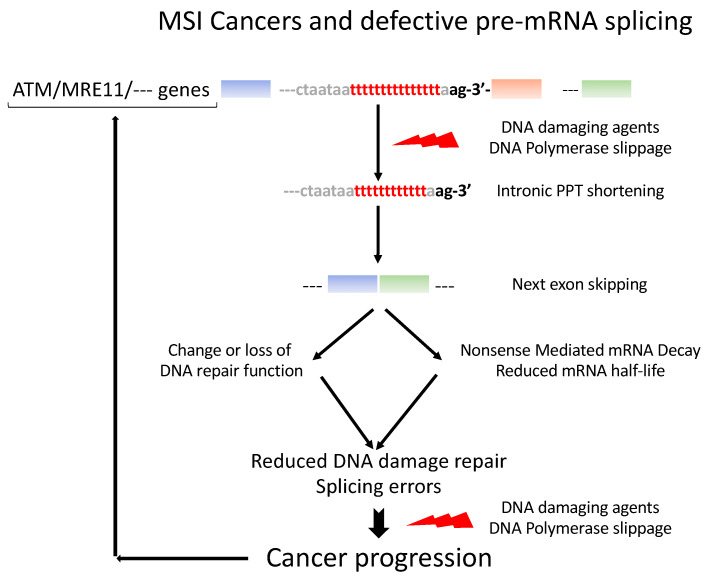
Model of incidence of pre-mRNA splicing errors in MSI cancer progression. DNA damage and replication errors trigger exon skipping in genes with intronic PPT shortening. The corresponding pre-mRNA structures are changed such that the function of the gene may be changed or the mRNA be targeted to the NMD surveillance system and degraded. The same errors may then be produced over again over the course of the cancer progression.

## 6. Conclusions

Much knowledge has been acquired with respect to pre-mRNA splicing mechanisms—although there is still a lot to discover for MSI cancers—to the point that restoring canonical splicing in cancers is no longer unrealistic. Indeed, anticancer trials that aim to evaluate the effects of drugs targeting several types of splicing defects are ongoing (registered at NIH ClinicalTrials.gouv). As such, MSI cancers that carry intronic PPT mutations should be eligible to be used in several corrective approaches. Pre-mRNA *trans*-splicing, which would replace the PPT-shortened transcript, would be an option, although somewhat difficult to use in vivo. Evidently, the use of the CrispR/cas9 approach would seem to be more appropriate. In addition, small molecule libraries could be screened to look for phenotype corrections, for example, through the use of straightforward methods that make use of marker-bearing minigenes.

## Data Availability

No new data were produced here. Only data that were extracted from the TCGA database and presented in Table 2, were produced in this article.

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
