# Peer review of "Microsatellite Instability and Aberrant Pre-mRNA Splicing: How Intimate Is It?"

_genes, 2023, doi:10.3390/genes14020311_

Round 1

Reviewer 1 Report

The review provides a clear update of current knowledge of alternative splicing and how dysregulation can lead to microsatellite instability. I found the review very simple to understand and read and it provides important context and insight into how MSI can develop and cause cancer (with the focus on CRC).  

As a general comment the grammar throughout needs to be double checked. There are often incorrect words used and/or in the wrong context, as though the article has been poorly translated. I have altered some text throughout (see pdf attached).

I also found the review to be almost too brief. The link between pre-mRNA spicing and MSI is a very topical content and feel there could be more a bit more detail in each section. For example:

o Alternative splicing as a therapeutic target or biomarker?

o Much is known about MSI in CRC. What about stomach and gastric cancers?

o How does alternative splicing affect cancer development versus progression? Are there examples of alternative splicing affecting other hallmarks of cancer (e.g. proliferation, apoptosis, angiogenesis, invasion/metastasis, drug resistance)?

o Are there any current therapies/ drugs that act to target the spliceosome or alternative splicing components? Can you give some examples of how these are proposed to work in CRC or other cancers?

As a final paragraph, there should include a general summary / Future perspectives / conclusion section to incorporate the current knowledge on pre-mRNA splicing  and perhaps insights into whether and how such events could act as biomarkers for disease response and/or whether these mechanisms can be targeted or manipulated for treatment of different diseases including cancer. ....is this possible or do we need to focus on some-how reactivating the downstream affected genes and DDR pathways.

Author Response

Responses to reviewer: We thank the reviewer for his input towards a better manuscript; language has been corrected according to requests throughout the manuscript.

1-Alternative splicing as a therapeutic target or biomarker? relevant text has been added, lanes 64-76. We believe that alternative splicing events can be both markers of cell states, and in the case of the SF3B1 (from the U2 spliceosome subunit), also an authentic target (see clinical trial in reference 3)

2-Much is known about MSI in What about stomach and gastric cancers? relevant sentences added lines 99 (ref. 30) and 110 (ref. 31)

3-How does alternative splicing affect cancer development versus progression? relevant sentence added lines 56,  57 (ref. 11); we showed that cancer development from polyps, in the case of the colon, was associated with numerous splicing alterations (ref. 11), suggesting that alternative splicing might contribute to cancer progression. We have not, as yet, evidence that altered splicing may be at the origin of cancer.

4-Are there examples of alternative splicing affecting other hallmarks of cancer (e.g. proliferation, apoptosis, angiogenesis, invasion/metastasis, drug resistance)? relevant sentences same as in response to comment 1 (BcLx splicing and apoptosis dependency). As to drug resistance, sentences lanes 46-48 (ref. 5, 6) were included to show that altered splicing was induced by various drugs that interfere with activity of the splicing machinery.

5-Are there any current therapies/ drugs that act to target the spliceosome or alternative splicing components? Can you give some examples of how these are proposed to work in CRC or other cancers? Relevant examples are described in references 21-23, including the ability of selected drugs, mainly directed against SF3B1, like E7107, an SF3B1 inhibitor derived from pladienolide B, to disrupt canonical splicing (lanes 72-76)

6- A final paragraph (Conclusions) has been added (lanes 217-226)

Reviewer 2 Report

The minireview/ hypothesis paper by Corcos et al. is generally well written and informative. I found only few minor issues.

II-Microsatellite Instability cancers and pre-mRNA splicing alterations (Table 1) (Figure 2) Microsatellite (MS) sequences, among whose repeated mono- or di-nucleotide sequences, occur throughout the genome, with a higher proportion in introns than in exons [15-17]. 

>A complicated sentence, please modify.

"slippage of the DNA polymerase, despite its proof-reading activity"

>DNA polymerase slippage is independent of its proof-reading activity. In slippage, a correct nucleotide has been inserted but the sequence prior to it has slipped.

Figure 2: Not sure how the “replication error” in this figure relates to the DNA pol slippage discussed earlier? It would be clearer to draw the slippage event and subsequent replication that results in lengthening or shortening of the nucleotide run.

Author Response

Comments and Suggestions for Authors: we thank the reviewer for his relevant observations

”II-Microsatellite Instability cancers and pre-mRNA splicing alterations (Table 1) (Figure 2) Microsatellite (MS) sequences, among whose repeated mono- or di-nucleotide sequences, occur throughout the genome, with a higher proportion in introns than in exons [15-17]. ”

>A complicated sentence, please modify.     The sentence has been modified (lanes 85-87)

"slippage of the DNA polymerase, despite its proof-reading activity"

>DNA polymerase slippage is independent of its proof-reading activity. In slippage, a correct nucleotide has been inserted but the sequence prior to it has slipped.         We removed the sentence mixing proof reading activity of DNA polymerase and slippage to retain only the text pertaining to slippage (lanes 87-89).

Figure 2: Not sure how the “replication error” in this figure relates to the DNA pol slippage discussed earlier? It would be clearer to draw the slippage event and subsequent replication that results in lengthening or shortening of the nucleotide run. Figure 2 has been changed